# Production of Enriched Biomass by Carotenogenic Yeasts Cultivated on by-Products of Poultry Processing—A Screening Study

**DOI:** 10.3390/microorganisms11020321

**Published:** 2023-01-27

**Authors:** Jiří Holub, Martin Szotkowski, Oleg Chujanov, Dominika Špačková, Pavlína Sniegoňová, Ivana Márová

**Affiliations:** Faculty of Chemistry, Brno University of Technology, Purkyňova 464/118, 612 00 Brno, Czech Republic

**Keywords:** carotenogenic yeasts, poultry waste fat, poultry feather, enriched biomass, carotenoids, lipids, biorefinery

## Abstract

Carotenogenic yeasts are a group of microorganisms producing valuable metabolites such as carotenoids, ergosterol, ubiquinone or fatty acids. Their exceptional adaptability allows them to grow in diverse conditions. Owing to their extracellular lipase activity, they are capable of processing many lipid-type waste substrates. This study discusses the processing of poultry waste, specifically fat and feathers by using carotenogenic yeasts. Poultry fat does not require any pre-treatment to be utilized by yeast, but hydrolytic pre-treatment is required for the utilization of the nitrogen contained in feathers. Glycerol was used as a supplementary substrate to support the culture in the early stages of growth. Seven yeast strains were used for the experiments, of which the strain *Rhodotorula mucilaginosa* CCY19-4-25 achieved exceptional results of biomass production: 29.5 g/L on poultry fat + 10% glycerol at C/N ratio 25 and 28.3 g/L on media containing poultry fat + 25% glycerol at C/N 50. The bioreactor cultivation of the *Rhodosporidium toruloides* strain in media containing glycerol and feather hydrolysate as a nitrogen substrate achieved a biomass yield of 34.92 g/L after 144 h of cultivation. The produced enriched yeast biomass can be used as a component for poultry feeding; thus, the study is performed under the biorefinery concept.

## 1. Introduction

Red yeasts are a group of diverse yeast species (*Rhodotorula* sp., *Sporidiobalus* sp., *Cystofilobasidium* sp.) capable of carotenoid biosynthesis via the mevalonate pathway [1,2]. These yeasts also include oleogenous species that are capable of accumulating lipids within their cells in a total content of more than 20% of dry biomass [3,4]. In many cases, carotenogenic yeasts possess extracellular lipase activity, which results in their ability to decompose triglycerides into glycerol and fatty acids and use them as a carbon source for the propagation and production of metabolites [5,6].

Carotenogenic yeasts are considered to be a species of interest due to their ability to biotransform waste substrates of a lipid, sugar or polyol (glycerol) nature (which has been observed in many studies [7,8,9,10,11,12]) to valuable metabolites, which include coenzyme Q10, ergosterol, carotenoids, fatty acids (lipids) and β-glucans [13].

The growing world population increases the demands on all sectors of industrial and agricultural production, the burden on the environment is being increased, and as a result it is necessary to intensify the transition from a linear economic model to a circular economic model [14,15]. Carotenogenic yeasts fit into the circular economic model with their metabolic activity, which enables the production of biomass and metabolites from waste organic substrates under the biorefinery concept. Furthermore, it is a significant advantage that the biotechnological production of biomass can be carried out in any place equipped with a suitable technology base, independently of the agricultural ground or the climate conditions [12]. The produced biomass enriched with beneficial substances (carotenoids, ergosterol, ubiquinone, lipids) has great potential as a feed additive, and at the same time, it is a cheaper alternative to complex carotenoid mixtures used as a feed additive [16,17]. According to a recent study, animals supplemented with enriched carotenogenic yeast biomass achieved better body growth and condition than animals whose diet was not supplemented with yeast [18]. Biomass or its components have the potential to be used in supplements for human nutrition, cosmetics, or pharmacy [19]. Feeding of poultry with red yeast biomass cultivated on waste poultry fat as the only carbon source and poultry feather hydrolysate as the only nitrogen source fully accepts the conception of the circular economy.

According to [20], the annual production of meat is 337 million tons, and more than 1/3 belongs to the production of chicken meat (118.02 million tons). Such mass production produces several usable by-products, including poultry feathers (40 million tons per year [21]) and fat (11% of total chicken body fat—not all waste by-products [22]). Poultry fat is rich in fatty acids (30.7% SFA, 30.2% MUFA and 30.9% PUFA) and can be considered a valuable carbon source [23]. As a source of nitrogen, it is possible to consider using feathers, the composition of which is 91% protein (keratin), 1% lipids and 8% water [24]. Due to the rigid structure of keratin, in which protein chains are tightly packed and stabilized through hydrophobic interactions and disulfide bonds, degradation by common proteolytic enzymes such as pepsin, papain or trypsin is unattainable [25].

By processing poultry material, a series of waste by-products can be obtained, which, after appropriate treatment, could be considered a carbonaceous or even complex substrate for biotechnological processing. The keratin contained in feathers is classified as a scleroprotein and due to its insolubility in water, it is difficult to utilize it in its native form by microorganisms without any specific enzymatic apparatus. For this reason, its basic or basic-enzymatic hydrolysis pre-treatment is necessary for its use in the cultivation of carotenogenic yeasts [26]. Under normal conditions, chicken fat is in a liquid form and there is no need for its pre-treatment as carotenogenic yeasts contain enzyme equipment (extracellular lipases) for splitting triglycerides.

## 2. Materials and Methods

### 2.1. Strains

The following types of red (carotenogenic) yeasts from the Bratislava collection of CCY (Culture Collection of Yeasts) strains were used for cultivation: *Cystofilobasidium macerans* CCY 10-1-2, *Rhodotorula kratochvilovae* CCY 20-2-26, *Rhodosporidium toruloides* CCY 062-002-001, *Rhodotorula mucilaginosa* CCY 20-9-7, *Rhodotorula mucilaginosa* CCY 19-4-25, *Sporidiobolus metaroseus* CCY 19-6-20, and *Sporidiobolus pararoseus* CCY 19-9-6.

### 2.2. Waste Materials and Their Pre-Treatment

Animal waste materials such as chicken fat and poultry feathers were used for the cultivations in combination with purified glycerol (which also served as a model for cultivations on waste glycerol from biofuel production; further GLY). While chicken fat was used for cultivation purposes in its pure form without any hydrolyzing treatments, degreased feathers were subjected to basic hydrolysis. The process consisted of hydrolyzing 80 g of degreased feathers in 800 mL of 1 M NaOH solution in sterilizing bottles at 100 °C for 30 min. The determination of nitrogen in the final hydrolysate was carried out using the biuret method. According to the results of nitrogen determination, 80 mL of hydrolysate per liter of medium should be used for cultivation purposes.

### 2.3. Description of Cultivation Experiments

Cultivations were carried out in 250 mL Erlenmeyer flasks with a medium content of 50 mL. The media contained a common mineral base (Table 1) and differed in the types and amounts of carbonaceous (chicken fat and glycerol; further F and GLY) and nitrogenous substrates (urea and feather hydrolysate—further as F.H.) used. Cultivation experiments were performed at C/N ratios of 25 and 50 and divided into series: the 1st series without the addition of hydrolysate and the 2nd series with the addition of hydrolysate. Both series included a control media containing glycerol as a carbon source, one containing urea as a nitrogen source and the second with addition of feather hydrolysate as a nitrogen source (Table 2 and Table 3).

### 2.4. Cultivation in Flasks

Yeast cultivation was carried out via double inoculation in Erlenmeyer flasks in YPD media (Table 4), according to an inoculation ratio of 1:5, on reciprocal shakers ensuring constant shaking. The double inoculation process was used based on experience and previously published results [12]. This cultivation scheme is sufficient for production of exponential culture with high cell density. Each inoculum was re-inoculated 24 h after first inoculation. Inoculation into the production media from the 2nd inoculum was performed at an inoculation ratio of 1:5 to 50 mL of the production media in 250 mL Erlenmeyer flasks. Production cultivation lasted 96 h at constant shaking on reciprocating shakers. Cultivation was terminated by separating the biomass from the medium by repeated centrifugation and subsequent freezing of the biomass at −80 °C to prepare it for the lyophilization procedure.

### 2.5. Cultivation in a Bioreactor

Cultivation was carried out in a 3.5 L bioreactor with a working volume of 2.2 L (2 L of production medium and 200 mL of inoculum). The bioreactor was filled with the production medium (Table 5). The sterility of the process was achieved by subjecting the bioreactor to sterilizing conditions in an autoclave (121 °C for 15 min). After cooling and connecting the bioreactor to the control unit, the pH was adjusted by using 10% (m/m) KOH and 10% (*v*/*v*) H_2_SO_4_ solutions to a working pH of 5.5. The temperature was set to 25 °C and constant stirring and aeration was ensured.

### 2.6. Gravimetry and Pigment Extraction

The frozen biomass was lyophilized and subjected to the gravimetric determination of biomass production (g/L). The hydrated biomass in microtubes filled with glass beads and methanol was subsequently subjected to a disintegration process using a laboratory disintegrator for the purpose of extracting pigments, ergosterol and ubiquinone according to Folch‘s extraction [27]. The final extract was stripped of the extraction solvent (chloroform) and was dissolved in a 2:1 HPLC grade solvent mixture of ethyl acetate:acetonitrile in a volume suitable for HPLC analysis with PDA detection [11,12].

### 2.7. Transesterification and FAME Extraction

The biomass (approx. 10 mg) was weighed into crimping vials filled with 1.8 mL of the transesterification mixture, which consisted of 15% (*v*/*v*) H_2_SO_4_ dissolved in methanol with the addition of heptadecenoic acid as an internal standard at a concentration of 0.5 mg/mL. The vials were crimped and tempered in a thermo-block at 85 °C for 120 min. The contents of the cooled vials were transferred to screw-top vials containing 1 mL of HPLC-grade hexane as an extractant and 0.5 mL of a neutralizing solution of 0.05 M NaOH. The mixture was intensively vortexed, and a proportional part (depending on the weight of the biomass) was taken from the hexane phase for the quantitative and qualitative analysis of fatty acids on a GC-FID apparatus.

### 2.8. HPLC Analysis

The qualitative and quantitative analysis of pigments, ergosterol and ubiquinone was performed on a Thermo Fischer Scientific HPLC apparatus on a Kinetex EVO C18 column (particle size 2.6 µm, length 150 mm, Phenomenex) with PDA detection and gradient elution. The flow rate was set at 1.2 mL/min with a total sample analysis duration of 25 min. The scanned wavelength channels were set to 450 nm for the detection of carotenoids and 280 nm for the detection of sterols and ubiquinone. The composition and gradient dosing of the mobile phases are introduced in Table 6 and Table 7.

### 2.9. GC Analysis

The analysis of fatty acids was carried out on the following apparatus: a Thermo Scientific TRACE 1300 TM Gas Chromatograph, with a Thermo Scientific Al 1310 autosampler and an automatic dispenser equipped with a splitter. FAME (fatty acid methyl ester) separation was performed on a LION GC-FAME column with dimensions of 30 m × 0.25 mm × 0.25 µm and detection was performed on a flame ionization detector (FID). The analysis conditions were as follows:

The temperature of the Injector was set to 240 °C, the split was set to 10, the flow rate of the mobile phases was 1 mL/min, and the injection volume was 1 µL. The temperature of the detector was set to 240 °C with the following set flow rates: air 350 mL/min, hydrogen 35 mL/min and nitrogen (makeup gas) 30 mL/min. The temperature gradient during separation was as follows: 80 °C from injection to t_R_ = 1 min, temperature increase to 140 °C with a gradient of 15 °C/min to t_R_ = 5 min, temperature increase to 190 °C with a gradient of 3 °C/min up to t_R_ = 21.7 min, increase with a temperature gradient of 25 °C/min to 260 °C with maintenance of this temperature for 1 min until the end of separation at t_R_ = 25.5 min [3,4].

## 3. Results

As a model red yeast with previously verified high production properties, the strain *Rhodotosporidium toruloides* (CCY 062-002-001) was tested first. The cultivation was performed in Erlenmeyer flasks at C/N 25 and C/N 50 and in a laboratory fermenter as well. The results are expressed in the form of graphs. The other strains were tested in Erlenmeyer flasks at C/N 25 and C/N 50 only. These results are documented in Appendix A. The strain R. mucilaginosa CCY 19-4-25 was found to be the best producer of biomass from the poultry wastes; thus, there results are also presented in some graphs. Experimental data are introduced in Appendix A.

### 3.1. Flask Cultivations of Strain Rhodotosporidium toruloides (CCY 062-002-001)

#### 3.1.1. Results for C/N 25

The cultivations on media of the first series showed the highest biomass growths in GLY (19.0 g/L) and FAT (20.4 g/L) media. Furthermore, two inverse trends can be observed in media containing a mixed carbonaceous substrates. In the first series, as the concentration of fat in the medium increases, the biomass production increases; in the second series, the trend is inverse. Significant carotenoid productions were achieved on GLY (4.891 mg/g) and F + 10% GLY (5.667 mg/g) media. The highest ergosterol productions achieved were observed on GLY (F.H.) (5.544 mg/g) and FAT + F.H. (6.911 mg/g) media. The highest accumulations of ubiquinone were achieved on F + 25% GLY (7.748 mg/g) and F + 10% GLY + F.H. (9.711 mg/g) media. A higher percentage content of lipids in the biomass was recorded in the mixed media of the second series, in which the highest accumulation of lipids in the dry biomass was recorded in F + 10% GLY + F.H. (25.04%). The highest productions of SFA (1.57 g/L medium) and MUFA (2.01 g/L medium) were observed in the FAT medium. The highest PUFA production (0.85 g/L medium) was recorded in the F + 25% GLY + F.H. medium. Results of cultivations are presented in Figure 1, Figure 2 and Figure 3.

#### 3.1.2. Results for C/N 50

The most significant biomass productions were achieved on the glycerol media GLY + F.H. (21 g/L) and GLY (24 g/L). In media with mixed carbonaceous substrates, there were observable trends in both series, in which the biomass production increased as the concentration of chicken fat in the medium decreased. The highest values of produced carotenoids were achieved in GLY (2.533 mg/g) and F + 25% GLY (4.582 mg/g) media. Significant ergosterol productions were achieved on F + 25% GLY + F.H. (8.724 mg/g) and F + 10% GLY + F.H. (8.944 mg/g) media. Significant values of ubiquinone production were achieved in F + 25% GLY + F.H. (9.706 mg/g) and GLY + F.H. (F.H) (11.611 mg/g) media. The highest percentage content of lipids in the biomass was achieved during cultivation on the F + 10% GLY (31.14%) medium. The highest production of individual types of fatty acids was observed in cultivations in different media: for SFA, it was on GLY + F.H. (2.09 g/L medium); for MUFA, it was on GLY (2.29 g/L medium); and for PUFA, it was on F + 25% GLY (0.81 g/L medium). In the second series, in media with a mixed carbon substrate, a trend of decreasing lipid accumulation can be observed depending on the decreasing concentration of fat in the media. Results of cultivations are presented in Figure 4, Figure 5 and Figure 6.

#### 3.1.3. Bioreactor Cultivation of Strain *Rhodotosporidium toruloides* (CCY 62-2-4)

The cultivation in a laboratory bioreactor showed that under controlled conditions, a significant growth in biomass can be achieved, as can be observed in Figure 7 in the 144th hour of cultivation (34.92 g/L). The production of carotenoids did not report significant increments; its maximum was reached at the 22nd hour of cultivation (0.831 mg/g). The concentration of ergosterol in the biomass reached its maximum at the 28th hour of cultivation (8.280 mg/g). The highest observed production of ubiquinone was at the beginning of cultivation (20.942 mg/g), but significant production was also achieved in the culture at the 144th hour of cultivation (12.059 mg/g). The highest percentage content of lipids in the biomass was observed at the 72nd hour of cultivation (40.81%), but the most significant production of all types of fatty acids was achieved at the 144th hour of cultivation (SFA 4.50 g/L, MUFA 5.15 g/L, PUFA 2.08 g/L). Results of cultivation are presented in Figure 7, Figure 8 and Figure 9.

### 3.2. Flask Cultivations of Strain Rhodotorula kratochvilovae (CCY 20-2-26)

#### 3.2.1. Results for C/N 25

The highest biomass production in the first series was recorded in mixed carbon substrate media, F + 10% GLY (17.7 g/L) and F + 25% GLY (19.2 g/L). Significant productions of carotenoids and ergosterol were achieved in the control media of the first series; in the GLY medium, the production of carotenoids reached 1.602 mg/g and ergosterol reached 3.443 mg/g of biomass, while in GLY + F.H., the production of carotenoids was 1.439 mg/g and ergosterol achieved a biomass of 3.778 mg/g. The highest ubiquinone productions were observed in the GLY medium (10.495 mg/g) and in the F + 25% GLY + F.H. medium (6.238 mg/g). A significant lipid production was recorded in the FAT + F.H. medium (40.82% of dry biomass), which generally showed the highest production of all types of fatty acids (SFA 2.26 g/L, MUFA 2.18 g/L and PUFA 1.31 g/L medium). There is also an observable trend in the second series of samples with F.H., in which with the increasing concentrations of fat in the media, the percentage accumulation of lipids in the biomass also increases (Appendix A).

#### 3.2.2. Results for C/N 50

In the first series with chicken fat, a significant biomass growth was observed, the highest biomass yield was achieved on F + 25% GLY (21.2 g/L) media. In the second series, with addition of fat, there is a trend of increasing growth of biomass with a decreasing content of fat in the media. The highest overall biomass concentration of both series was achieved in the media consisting of F + 25% GLY + F.H. (21.5 g/L). The production of carotenoids on the mixed media did not exceed the increments in the control glycerol media, GLY (1.196 mg/g) and GLY (F.H.) (1.324 mg/g). The increased productions of ergosterol were observed in F + 10% GLY + F.H. (11.583 mg/g) media. The highest ubiquinone productions were recorded on the control media of the second series, GLY (F.H.) (13.149 mg/g) and GLY + F.H. (F.H) (15.505 mg/g). Lipid accumulation achieved the greatest values in the cultivation of *Rhodotorula kratochvilovae* in F + 25% GLY + F.H. (37.87% dry biomass) together with the highest productions of all types of fatty acids (SFA 2.09 g/L, MUFA 3.90 g/L and PUFA 2.15 g/L medium).

### 3.3. Flask Cultivations of Strain Cystofilobasidium macerans (CCY 10-1-2)

#### 3.3.1. Results of C/N Ratio 25

In cultivations in media with a mixed carbon source, there is an observable trend of increasing biomass production with a decreasing fat concentration in the medium. The highest yields of products were achieved in F + 25% GLY + F.H. (15.1 g/L) and GLY + F.H. (F.H.) (16.6 g/L) media. The notable carotenoid productions were observed in GLY + F.H. (1.325 mg/g) and GLY (1.963 mg/g) media. The highest reported results of ergosterol production were achieved on F + 25% GLY + F.H. (6.587 mg/g) and FAT + F.H. (9.932 mg/g) media. Noteworthy values of production of ubiquinone were achieved in GLY (10.939 mg/g) and F + 25% GLY + F.H. (7.951 mg/g) media. The highest values of accumulated lipids by biomass (31.88%) and produced SFA (2.31 g/L) were observed after cultivation in FAT + F.H. media. The highest accumulation of MUFA by the strain was achieved in GLY + F.H. (F.H.) (1.41 g/L) and PUFA reached its highest value in F + 25% GLY + F.H. (1.02 g/L). The percentage content of lipids in biomass reported two reverse trends in media with a mixed carbon source. In the first series, with an increasing level of fat, the amount of accumulated lipids decreased. This trend is inverse in the second series (Appendix A).

#### 3.3.2. Results for C/N 50

The most significant biomass growth of the yeast *Cystofilobasidium macerans* at a C/N ratio of 50 was achieved in GLY + F.H. (F.H.) (17.5 g/L) and GLY (F.H.) (21.3 g/L) media. In media with mixed carbon substrates of the first series, an increasing trend of biomass production can be observed depending on the decrease in fat concentration in the media composition. Significant carotenoid productions were observed in cultivation in FAT (4.916 mg/g) and F + 10% GLY (4.056 mg/g) media. The highest ergosterol products were achieved in F + 25% GLY + F.H. (6.456 mg/g) and F + 10% GLY + F.H. (8.72 mg/g) media. The highest amount of ubiquinone was achieved in GLY (F.H.) (8.228 mg/g) and GLY + F.H. (F.H) (9.864 mg/g) media. The notable percentage content of lipids in dry biomass was observed after cultivation in the FAT + F.H. (26.46%) medium. The most significant production of SFA (1.73 g/L) and MUFA (2.80 g/L) were recorded in GLY (F.H.) media. PUFA were produced to the greatest extent in F + 25% GLY + F.H. (0.83 g/L) media.

### 3.4. Flask Cultivation of Strain Rhodotorula mucilaginosa (CCY 19-4-25)

#### 3.4.1. Results of C/N Ratio 25

The trend of decreasing biomass production depending on the decreasing concentration of fat in the media can be observed in media with a mixed carbon substrate of the first series. The highest biomass yields were achieved in F + 10% GLY (24.8 g/L) and FAT (29.5 g/L) media. Significant carotenoid productions were achieved in F + 25% GLY (7.430 mg/g) and GLY (10,469 mg/g) media, together with highest ergosterol productions of 5.121 mg/g in F + 25% GLY and 6.832 mg/g in GLY media. Ubiquinone was produced notably in media GLY (5.726 mg/g) and FAT + F.H. (8.932 mg/g). The highest lipid accumulation (25% dry biomass) and production of SFA (3.0 g/L), MUFA (3.12 g/L) and PUFA (1.39 g/L) were observed on the FAT medium. The lipid content percentage of the mixed carbon source media in the second series reported an increasing tendency with the decreasing fat content of the media (Appendix A). Results of cultivations are also presented in Figure 10, Figure 11 and Figure 12.

#### 3.4.2. Results of C/N Ratio 50

The most significant biomass productions were achieved in FAT + F.H. (28.3 g/L) and F + 25% GLY + F.H. (28.3 g/L) media. Significant carotenoid productions were observed in cultures in F + 25% GLY + F.H. (5.612 mg/g) and GLY (F.H.) (6.044 mg/g) media. The highest concentrations of ergosterol were achieved in GLY (F.H.) (6.118 mg/g) and GLY + F.H. (F.H.) (4.621 mg/g) media. Ubiquinone was produced in the highest concentrations in F + 25% GLY + F.H. (7.573 mg/g) and FAT + F.H. (7.838 mg/g) media. The percentage of lipid content (38.33%) and SFA production (5.76 g/L) was observed in the culture in the F + 10% GLY + F.H. medium. The highest production of MUFA (3.43 g/L) and PUFA (2.43 g/L) was observed in the FAT + F.H. medium. Results of cultivations are presented in Figure 13, Figure 14 and Figure 15.

### 3.5. Flask Cultivations of Strain Rhodotorula mucilaginosa (CCY 20-9-7)

#### 3.5.1. Results of C/N Ratio 25

Biomass production reached its two highest values in F + 25% GLY + F.H. (17.2 g/L) and GLY (F.H.) (17.8 g/L) media. The highest concentrations of carotenoids in the biomass were observed in cultivations in F + 10% GLY (1.324 mg/g) and GLY (2.241 mg/g) media. Ergosterol was observed in high concentrations in the biomass cultured in GLY (F.H.) (4.945 mg/g) and GLY + F.H. (F.H.) (4.969 mg/g) media. The highest ubiquinone productions were achieved in FAT + F.H. (5.418 mg/g) and GLY + F.H. (F.H.) (5.988 mg/g) media. The most significant lipid accumulation was achieved in the FAT medium (46.79%), but the highest production of SFA (2.97 g/L) was observed in the GLY (F.H.) medium and the highest production of MUFA (1.24 g/L) and PUFA (1.22 g/L) was recorded in cultivation in the F + 10% GLY + F.H. medium (Appendix A).

#### 3.5.2. Results of C/N Ratio 50

Significant biomass production values were achieved in F + 25% GLY + F.H. (24.8 g/L) and FAT + F.H. (26.1 g/L) media. The highest values of accumulated carotenoid concentrations were achieved in cultivation in GLY (1.187 mg/g) and F + 10% GLY (1.221 mg/g) media. Ergosterol was produced in greater quantities in FAT + F.H. (3.895 mg/g) and F + 10% GLY + F.H. (3.704 mg/g) media. Ubiquinone was produced on a greater scale in F + 10% GLY (6.084 mg/g) and FAT + F.H. (7.894 mg/g) media. A slightly increasing trend of lipid accumulation, depending on the decreasing concentration of fat in the media, is observable in the second series in media with mixed carbonaceous substrates. The highest accumulation of lipids (34.59%) and the highest production of SFA (4.66 g/L) were observed in the F + 25% GLY + F.H. medium. The highest production of MUFA (2.88 g/L) was recorded in the FAT + F.H. medium and the highest PUFA production (1.91 g/L) was observed in cultivation in the F + 10% GLY + F.H. medium.

### 3.6. Flask Cultivation of Strain Sporidiobolus metaroseus (CCY 19-6-20)

#### 3.6.1. Results for C/N Ratio 25

The highest biomass productions were observed in the second series of cultivations in F + 10% GLY + F.H. (14 g/L) and F + 25% GLY + F.H. (15.2 g/L) media. Carotenoids were produced by biomass to the greatest extent in GLY (F.H.) (1.247 mg/g) and GLY + F.H. (F.H.) (1.675 mg/g) media. Ergosterol reached the highest productions in GLY + F.H. (3.757 mg/g) and GLY (4.295 mg/g) media. The most significant ubiquinone productions were observed in GLY (F.H.) (5.490 mg/g) and F + 25% GLY + F.H. (5.610 mg/g) media. In media with mixed carbon substrates of the second series, a trend of increasing biomass production and lipid accumulation can be observed with decreasing fat concentration in the media. The highest percentage content of lipids in the biomass (36.41%) and the highest production of SFA (2.81 g/L) and MUFA (1.70 g/L) were observed in the F + 25% GLY + F.H. medium. The highest production of PUFA (1.17 g/L) was observed in the F + 10% GLY + F.H. medium (Appendix A).

#### 3.6.2. Results for C/N Ratio 50

The highest biomass productions were achieved in F + 10% GLY + F.H. (11 g/L) and F + 25% GLY + F.H. (15.7 g/L) media. The most notable carotenoid productions were observed in GLY (F.H.) (0.931 mg/g) and GLY + F.H. (F.H.) (1.255 g/L) media. The highest productions of ergosterol were measured in the same media, 2.810 mg/g in GLY (F.H.) and 3.754 mg/g in GLY + F.H. (F.H.). Ubiquinone productions were the highest in F + 25% GLY + F.H. (7.482 mg/g) and GLY + F.H. (F.H.) (7.845 mg/g) media. There is a noticeable trend of an increasing biomass growth and at the same time a decreasing percentage of lipids in the biomass, depending on the decreasing concentration of fat in the media. This phenomenon is observable in media with a mixed carbon substrate in the second series. The highest percentage of lipids was observed on the FAT + F.H. (44.23%) medium. The highest production of all types of fatty acids was observed in the F + 25% GLY + F.H. (SFA 1.18 g/L, MUFA 2.18 g/L, PUFA 1.66 g/L) medium.

### 3.7. Flask Cultivation of Strain Sporidiobolus pararoseus (CCY 19-9-6)

#### 3.7.1. Results of C/N Ratio 25

The most significant biomass growths were achieved in F + 25% GLY (15.9 g/L) and FAT + F.H. (16.1 g/L) media. The notable carotenoid productions were observed after cultivation in F + 10% GLY (1.845 mg/g) and GLY (2.690 mg/g) media. The concentrations of ergosterol in the biomass reached the highest values in GLY (F.H.) (8.231 mg/g) and GLY (9.247 mg/g) media. Ubiquinone was produced by biomass in the greatest values in FAT + F.H. (14.712 mg/g) and F + 10% GLY + F.H. (15.251 mg/g) media. In the second series of media with mixed carbonaceous substrates, a trend of decreasing lipid accumulation can be observed according to the decreasing fat content in the media. The highest accumulation of lipids (32.30%) and the production of SFA (2.48 g/L) and MUFA (1.85 g/L) were observed in cultivation in FAT + F.H. The highest production of PUFA (0.92 g/L) was observed in the F + 25% GLY + F.H. medium (Appendix A).

#### 3.7.2. Results of C/N 50

The most significant biomass productions were achieved in media of the second series, in F + 25% GLY + F.H. (28.2 g/L) and F + 10% GLY + F.H. (30.1 g/L). The culture achieved the highest production of carotenoids in GLY (F.H.) (1.496 mg/g) and F + 10% GLY + F.H. (2.425 mg/g) media. The highest values of ergosterol were achieved on GLY (7.181 mg/g) and GLY (F.H.) (14,365 mg/g) media. Significant ubiquinone productions were recorded in F + 25% GLY + F.H. (9.008 mg/g) and GLY (F.H.) (9.460 mg/g) media. In the first series, it can be observed that in media with a mixed carbon substrate, as the concentration of fat in the medium decreases, the production of biomass and accumulated lipids decrease slightly. The highest achieved percentage content of lipids in the biomass was observed in the FAT medium (35.13%). The highest production of all types of fatty acids was observed on the F + 10% GLY + F.H. (SFA 4.29 g/L, MUFA 4.04 g/L, PUFA 1.99 g/L) medium.

## 4. Discussion

Carotenogenic yeasts are ubiquitous microorganisms capable of utilizing various types of substrates and able to grow under various conditions. Their ability to accumulate lipids in higher amounts than 20% of the cell content places them in the group of oleaginous microorganisms. Many studies have confirmed their ability to utilize lipid substrates such as waste coffee oil, waste animal fat or waste frying oil [11,12]. According to these studies, this work focuses on the usability of poultry processing by-products, namely poultry fat and feathers, in media for the cultivation of carotenogenic yeasts. Carbon substrate was enriched with the addition of glycerol as a substrate that was easier to process in case of diauxia in order to provide to the yeast culture sufficient time to adapt to a more complex fatty substrate.

The data show that the significant producer of biomass at C/N 25 is the yeast *Rhodotorula mucilaginosa* (CCY 19-4-25), which showed the highest biomass growths in the FAT (29.5 g/L) and F + 10% GLY medium (24.8 g/L). At the same time the highest accumulations of carotenoids were recorded in GLY (10,469 mg/g) and F + 25% GLY (7.430 mg/g) media. The most significant accumulations of ergosterol were recorded in *Cystofilobasidium macerans* strains in the FAT + F.H. medium (9.932 mg/g) and *Sporidiobolus pararoseus* in the GLY medium (9.247 mg/g). The *Sporidiobolus pararoseus* strain showed the highest ubiquinone accumulation in F + 10% GLY + F.H. (15.251 mg/g) and FAT + F.H. (14.712 mg/g) media. The most accumulated lipids were recorded in the yeast culture *Rhodotorula mucilaginosa* (CCY 20-9-7) in the FAT medium (46.79%) and *Rhodotorula kratochvilovae* in the FAT + F.H. medium (40.82% with lipid production of 5.75 g/L). The highest lipid production was recorded in the yeast *Rhodotorula mucilaginosa* (CCY 19-4-25) in the FAT medium (7.52 g/L)

During cultivations at C/N 50, significant increases in biomass were observed in the yeast strain *Sporidiobolus pararoseus* in the F + 10% GLY + F.H. (30.1 g/L) medium and *Rhodotorula mucilaginosa* (CCY 19-4-25) in FAT + F.H. and F + 25% GLY + F.H. media (28.3 g/L). On the same strain, significant carotenoid production was also found in the GLY (F.H.) medium (6.044 mg/g) and F + 25% GLY + F.H. medium (5.613 mg/g). The most significant ergosterol productions were recorded in strains of *Sporidiobolus pararoseus* on the GLY (F.H.) medium (14.365 mg/g) and *Rhodotorula kratochvilovae* in the F + 10% GLY + F.H. medium (11,584 mg/g). In the *Rhodotorula kratochvilovae* strain, the most significant production of ubiquinone was observed in GLY + F.H. (F.H.) (15.505 mg/g) and GLY (F.H.) (13.149 mg/g) media. A significant accumulation of lipids and the percentage content of lipids in the biomass at the same time were observed in the strain *Rhodotorula mucilaginosa* (CCY 19-4-25) in the F + 10% GLY + F.H. medium (38.33% with a lipid production of 10.62 g/L). Another important producer of lipids was the strain *Rhodotorula kratochvilovae* on the F + 25% GLY + F.H. medium (with lipid accumulation 37.87%). The second most significant value of lipid production was recorded for the strain *Sporidiobolus pararoseus* in the F + 10% GLY + F.H. (10.32 g/L) medium.

The *Rhodosporidium toruloides* strain was selected for cultivation in the bioreactor. This strain was used as a comparative culture in the study [12] where the effect of the use of lipidic waste substrates on the growth of biomass in a bioreactor was tested. Similar conditions as in the study were induced in our bioreactor cultivation to provide the possibility to compare the behavior of the same genus of yeast on a different carbon and nitrogen substrate at the same C/N ratio. The study shows that the final biomass growth during cultivation in the bioreactor was 10.46 g/L after 96 h (the carbon source consisted of 25% hydrolysate from spent coffee grounds and 75% from coffee oil, with urea used as a nitrogen source). In the second case, where a mixture containing 25% of spent coffee ground hydrolysate and 75% of waste frying oil was used as a carbon source, the biomass growth was 11.75 g/L after 96 h of cultivation [12]. In the case of cultivation on glycerol and feather hydrolysate, the maximum growth in the biomass achieved at the same time interval as in the comparative study (0–96 h) was at the 72nd h of cultivation, with a growth of 20.51 g/L, which is 96.08% more compared to the first value and 74.55% more compared to the second value. The continuation of this cultivation up to 144th h resulted in a final biomass yield of 34.92 g/L. The better growth in our induced conditions may be explained by the easier utilization of glycerol by yeasts compared to the more complex sugar–lipid substrate. This experiment could serve as model for cultivation on F.H. with waste glycerol, which is by-product in biofuel processing.

The main goal of the work was to find out whether carotenogenic yeasts are capable of utilizing a complex substrate in the form of feather hydrolysate and poultry fat. This theory was confirmed, while very interesting results were achieved both from the point of view of the accumulation of intracellular metabolites and biomass growth, but also from the point of view of a new approach to these types of waste, which could be biotechnologically used to produce feed biomass, food, or food supplements using carotenogenic yeast cultivation in the future.

## 5. Conclusions

Feeding of poultry with red yeast biomass cultivated on waste poultry fat as the only carbon source and poultry feather hydrolysate as the only nitrogen source fully accepts the conception of the circular economy. Animals supplemented with enriched carotenogenic yeast biomass achieved better body growth and condition than animals whose diet was not supplemented with yeasts. Biomass or its components have the potential to be used not only in poultry biorefinery, but also in supplements for human nutrition, cosmetics, or pharmacy.

## Figures and Tables

**Figure 1 microorganisms-11-00321-f001:**
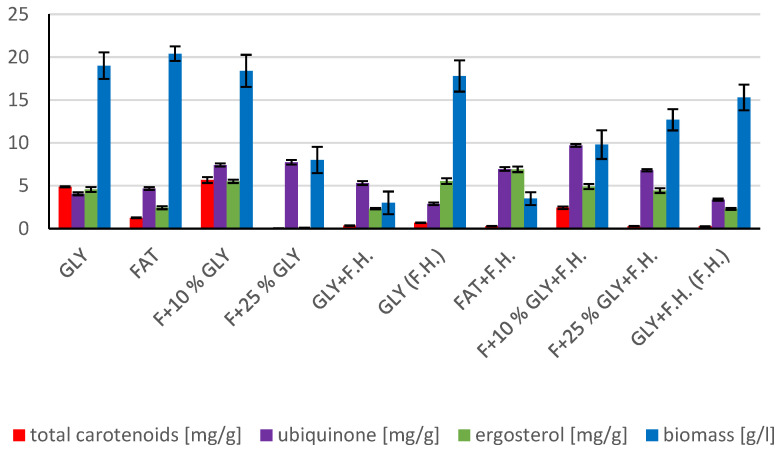
Metabolite production by biomass of yeast strain *Rhodotosporidium toruloides* in conditions of C/N ratio 25.

**Figure 2 microorganisms-11-00321-f002:**
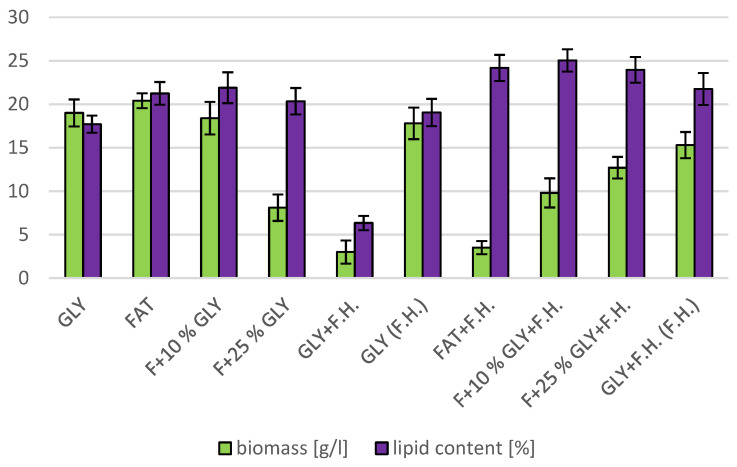
Lipid and biomass content in yeast strain *Rhodotosporidium toruloides* in conditions of C/N ratio 25.

**Figure 3 microorganisms-11-00321-f003:**
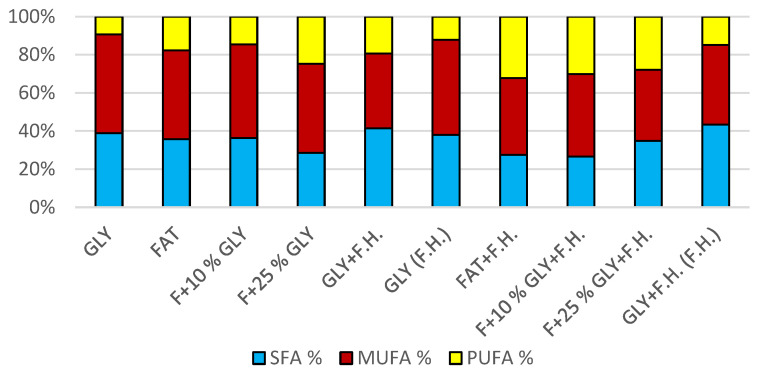
Fatty acid distribution in lipids of yeast strain *Rhodotosporidium toruloides* in conditions of C/N ratio 25.

**Figure 4 microorganisms-11-00321-f004:**
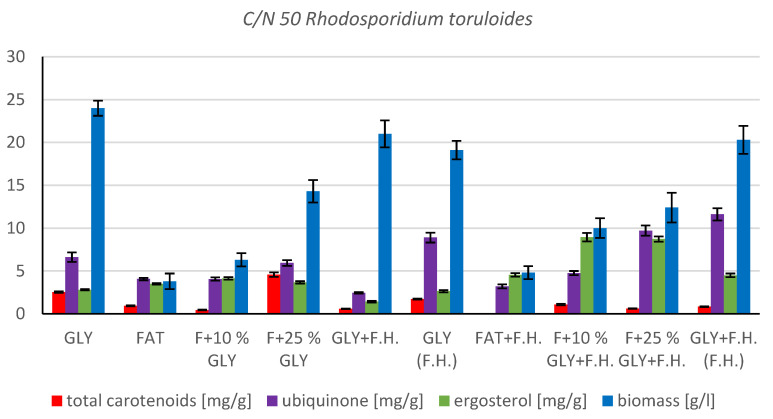
Metabolite production by biomass of yeast strain *Rhodotosporidium toruloides* in conditions of C/N ratio 50.

**Figure 5 microorganisms-11-00321-f005:**
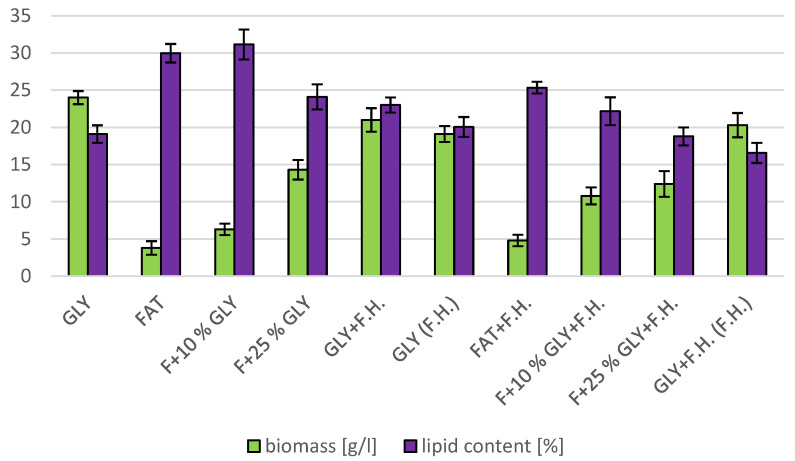
Lipid and biomass content in yeast strain *Rhodotosporidium toruloides* in conditions of C/N ratio 50.

**Figure 6 microorganisms-11-00321-f006:**
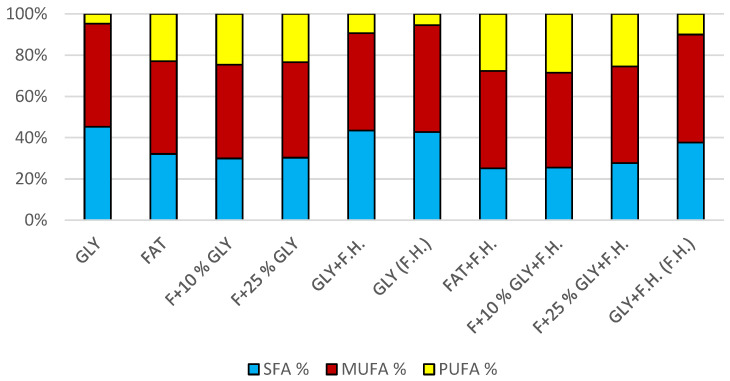
Fatty acid distribution in lipids of yeast strain *Rhodotosporidium toruloides* in conditions of C/N ratio 50.

**Figure 7 microorganisms-11-00321-f007:**
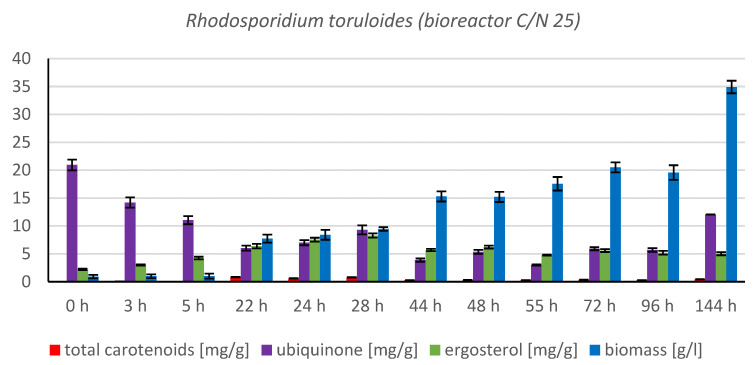
Metabolite production by biomass of yeast strain *Rhodotosporidium toruloides* in conditions of C/N ratio 25 cultivated in bioreactor.

**Figure 8 microorganisms-11-00321-f008:**
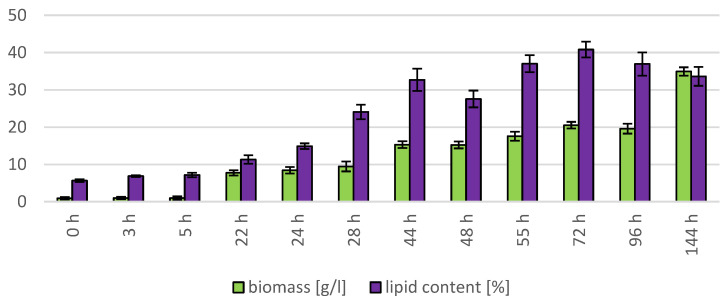
Lipid and biomass content in yeast strain *R. toruloides* in conditions of C/N ratio 25 cultivated in bioreactor.

**Figure 9 microorganisms-11-00321-f009:**
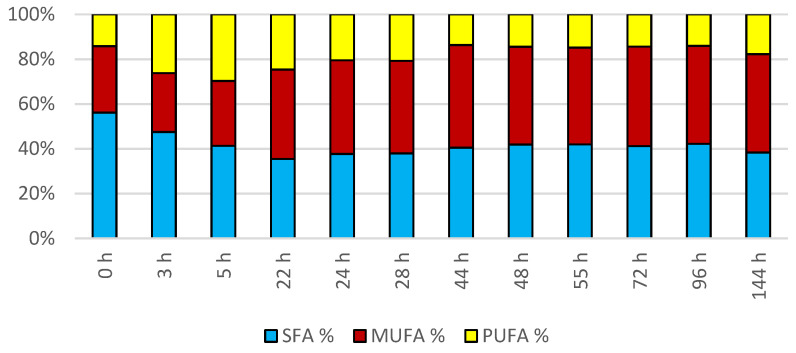
Fatty acid distribution in lipids of yeast strain *Rhodotosporidium toruloides* in conditions of C/N ratio 25 cultivated in bioreactor.

**Figure 10 microorganisms-11-00321-f010:**
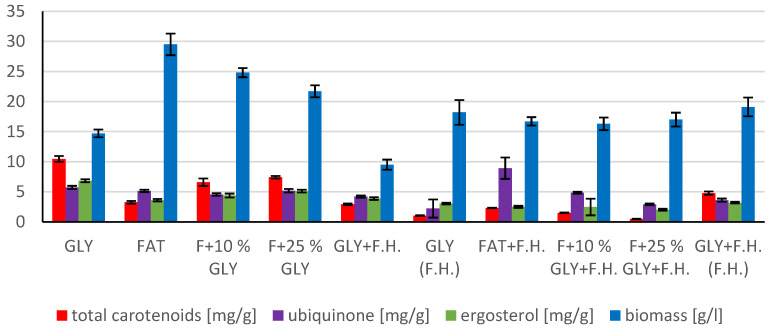
Metabolite production by biomass of yeast strain *Rhodotorula mucilaginosa* CCY19-4-25 in conditions of C/N ratio 25.

**Figure 11 microorganisms-11-00321-f011:**
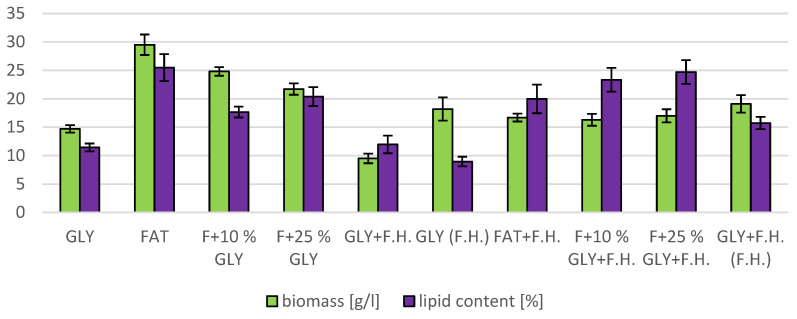
Lipid and biomass content in yeast strain *R. mucilaginosa* at C/N ratio 25.

**Figure 12 microorganisms-11-00321-f012:**
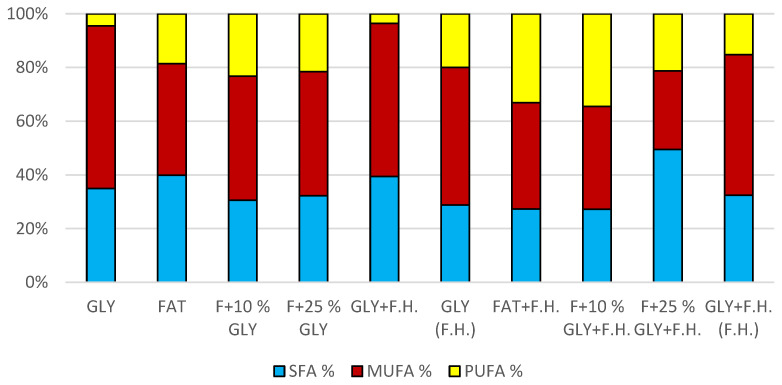
Fatty acid distribution in lipids of yeast strain *Rhodotorula mucilaginosa* in conditions of C/N ratio 25.

**Figure 13 microorganisms-11-00321-f013:**
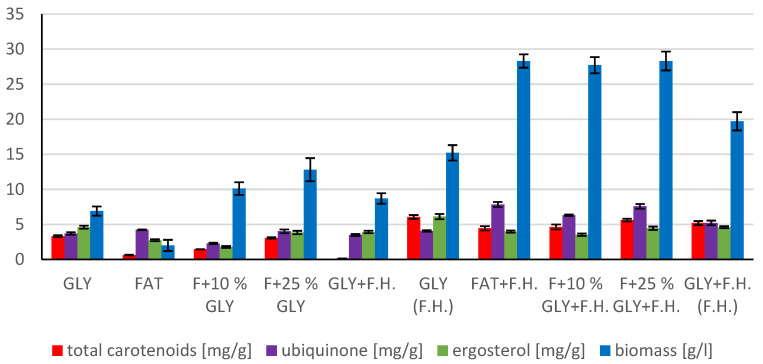
Metabolite production by biomass of yeast strain *Rhodotorula mucilaginosa* CCY19-4-25 in conditions of C/N ratio 50.

**Figure 14 microorganisms-11-00321-f014:**
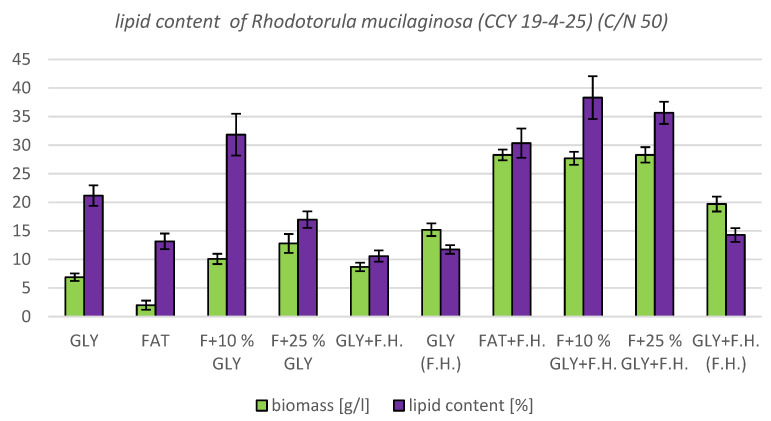
Biomass and lipid production by biomass of yeast strain *Rhodotorula mucilaginosa* (CCY 19-4-25) in conditions of C/N ratio 50.

**Figure 15 microorganisms-11-00321-f015:**
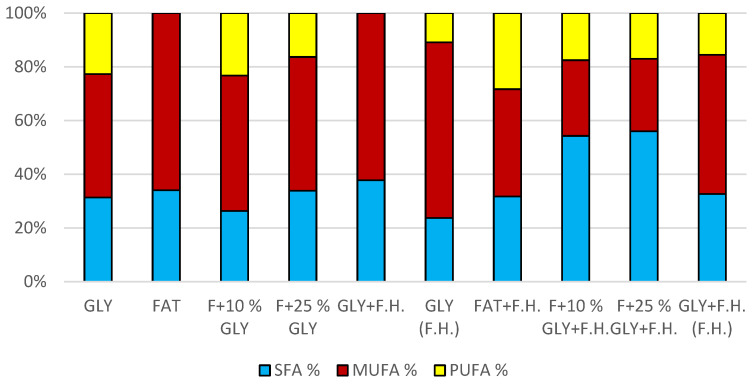
Fatty acid distribution in lipids of yeast strain *Rhodotorula mucilaginosa* in conditions of C/N ratio 50.

**Table 1 microorganisms-11-00321-t001:** Mineral basis of production media.

Component	Amount (g)
KH_2_PO_4_ (g)	4
MgSO_4_∙7H_2_O (g)	0.7
Tap water	1000

**Table 2 microorganisms-11-00321-t002:** Composition of production media at C/N ratio of 25 for volume of 1000 mL of media.

1st series					
Media name	GLY	FAT	F + 10% GLY	F + 25% GLY	GLY + F.H.
Poultry fat (g)	0	17.66	15.89	13.24	0
Glycerol (g)	46.26	0	4.63	11.56	46.26
Urea (g)	1.81	1.81	1.81	1.81	0
F.H. (mL)	0	0	0	0	80
2nd series					
Media name	GLY (−F.H.)	FAT + F.H.	F + 10% GLY + F.H.	F + 25% GLY + F.H.	GLY + F.H. (F.H)
Poultry fat (g)	0	17.66	15.89	13.24	0
Glycerol (g)	46.26	0	4.63	11.56	46.26
Urea (g)	1.81	0	0	0	0
F.H. (mL)	0	80	80	80	80

**Table 3 microorganisms-11-00321-t003:** Composition of production media at C/N ratio of 50 for volume of 1000 mL of media.

1st series					
Media name	GLY	FAT	F + 10% GLY	F + 25% GLY	GLY + F.H.
Poultry fat (g)	0	35.31	31.78	26.49	0
Glycerol (g)	92.51	0	9.25	23.13	92.51
Urea (g)	1.81	1.81	1.81	1.81	0
F.H. (mL)	0	0	0	0	80
2nd series					
Media name	GLY (F.H.)	FAT + F.H.	F + 10% GLY + F.H.	F + 25% GLY + F.H.	GLY + F.H. (F.H)
Poultry fat (g)	0	35.31	31.78	26.49	0
Glycerol (g)	92.51	0	9.25	23.13	92.51
Urea (g)	1.81	0	0	0	0
F.H. (mL)	0	80	80	80	80

**Table 4 microorganisms-11-00321-t004:** YPD media composition.

Component	Amount
Yeast autolysate (g)	10
Bacteriological peptone (g)	20
Glycerol (g)	20
Tap water (mL)	1000

**Table 5 microorganisms-11-00321-t005:** Composition of the bioreactor medium for volume of 2000 mL (the remaining volume was supplemented with tap water).

Component	Amount
KH_2_PO_4_ (g)	8
MgSO_4_∙7H_2_O (g)	1.392
Glycerol (g)	92.51
F.H. (mL)	160

**Table 6 microorganisms-11-00321-t006:** Composition of mobile phases in HPLC analysis.

Mobile Phase	Component	Volume Parts
A	Acetonitril	84
	100 mM trisHCl buffer pH 8	14
	Methanol	2
B	Methanol	60
	Ethylacetate	40

**Table 7 microorganisms-11-00321-t007:** Elution gradient.

Retention Time [min]	MF A (%)	MF B (%)
0	100	0
13	0	100
19	0	100
20	100	0
25	100	0

## Data Availability

No new data were created or analyzed in this study. Data sharing is not applicable to this article.

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
