# Peer review of "Production of Enriched Biomass by Carotenogenic Yeasts Cultivated on by-Products of Poultry Processing—A Screening Study"

_microorganisms, 2023, doi:10.3390/microorganisms11020321_

Round 1

Reviewer 1 Report

"Production of enriched biomass by carotenogenic yeasts cultivated on by-products of poultry processing – a screening study" written by Holub et al. dealt with the possibility of using residual biomasses such as poultry fat and feather as Carbon and Nitrogen sources to grow carotenogenic yeasts.  Seven different inoculum were investigated on a laboratory scale and one condition was scaled up in a 3.5 L bio-reactor. The manuscript looks to be appropriate to this journal,  but not in the present form. The following aspects should be addressed in order to make it worth of being published: - a more detailed characterization of the poultry fat used in the experiments should be reported: were they composed by glycerides or fatty acids (FA)? A detailed profile of FA should be reported, as well as the definition of SFA, MUFA and PUFA should be included. -   According to the present version, it was not possible to evaluate how the amounts of poultry fat (and Carbon) were defined and added. In addition, a more detailed study including metal profiles and target molecules monitored in this study should be evaluated. For sure, when the FH was used as a Nitrogen source, the carbon introduced by this waste was not considered and included in its definition. - Experiments results were reported as bar graphs without reporting any variability bars. Even in a screening study, the replicability of results and the relevant variability represent the first step to understanding the robustness of the experimentation. In the alternative, replicates of analysis on the centrifuged lyophilized biomass should be run.  - As a final remark, how the authors consider that residue recovered through centrifugation were carotenogenic yeasts? It was also useful to understand the native concentration of ergosterol, ubiquinone and carotenoids in the inoculum adopted to evaluate if the recovered ones were ponderously the same of that one introduced. In any case, analysis on the liquid removed through centrifugation should be necessary. What about the atom economy of the process? And what about the fate of these residues? Amounts of fatty acids, Nitrogen and phosphorous should be determined and a sort of efficacy of the fermentation evaluated.  

Author Response

Reviewer #1

RE: „ Jiří Holub et al.: „Production of enriched biomass by carotenogenic yeasts cultivated on by-products of poultry processing – a screening study.“   manuscript-revised.

Foremost, we would like to thank you for your great efforts in providing constructive criticism. We have carried out this revision according to the comments made by you as outlined below. All changes are indicated in BLUE within the manuscript. For these changes page or line numbers are also included.

Reviewer #1 Comments and Suggestions for Authors

A more detailed characterization of the poultry fat used in the experiments should be reported: were they composed by glycerides or fatty acids (FA)? A detailed profile of FA should be reported, as well as the definition of SFA, MUFA and PUFA should be included.

Changes we made in the manuscript:

Characterization of general poultry waste was found in publication [23]. Poultry fat waste used in our experiments differed. Its composition is described in supplementary file in Table S1..

Reviewer #1 Comments and Suggestions for Authors

According to the present version, it was not possible to evaluate how the amounts of poultry fat (and Carbon) were defined and added. In addition, a more detailed study including metal profiles and target molecules monitored in this study should be evaluated. For sure, when the FH was used as a Nitrogen source, the carbon introduced by this waste was not considered and included in its definition.

Our comments:

The amount of poultry waste fat used in production media in screening and bioreactor cultivations was calculated to match the amount of carbon  according to the control media. The amount used was based on the GC analysis of fatty acid profile and total triglyceride concentration. Due to the high C/N ratio and thus the high content of the carbon source, we neglected the amount of carbon delivered through the feather hydrolyzate. As a result, there could be only a small change in the C/N ratio (±2), which, however, from our experience and previous results, does not have a significant effect on changes in the production of both biomass and monitored metabolites. In our future experiments, we will incorporate this idea you suggested into the procedure to further refine the media composition and data obtained. Thank you

Reviewer #1  Comments and Suggestions for Authors

Experiments results were reported as bar graphs without reporting any variability bars. Even in a screening study, the replicability of results and the relevant variability represent the first step to understanding the robustness of the experimentation. In the alternative, replicates of analysis on the centrifuged lyophilized biomass should be run.

Our comments:  

Our screening culture experiments were performed in duplicate. In the manuscript, we redesigned the graphs and tables and added error bars that were not in the original version of the manuscript. Thank you for notifying us of this error.

Changes we made in the manuscript:

We added variability bars into graphs for biomass production, lipid content of biomass, and for total carotenoids, ergosterol and ubiquinone production. Examples of those graphs are listed below. The rest of graphs is included in manusctript.

Figure 1: Metabolite production by biomass of yeast strain Rhodotosporidium toruloides in conditions of C/N ratio 25

Figure 2: Lipid and biomass content in yeast strain Rhodotosporidium toruloides in conditions of C/N ratio 25.

Reviewer #1 Comments and Suggestions for Authors

As a final remark, how the authors consider that residue recovered through centrifugation were carotenogenic yeasts? It was also useful to understand the native concentration of ergosterol, ubiquinone and carotenoids in the inoculum adopted to evaluate if the recovered ones were ponderously the same of that one introduced. In any case, analysis on the liquid removed through centrifugation should be necessary. What about the atom economy of the process? And what about the fate of these residues? Amounts of fatty acids, Nitrogen and phosphorous should be determined and a sort of efficacy of the fermentation evaluated.  

Our comments:

In the cultivation process, in both the upstream and downstream processes, the culture is controlled at several points. The 1st point is the preparation of the inoculum itself. When we start from stock cultures delivered from the CCY collection in Bratislava, where the authenticity of the strain is regularly checked by genetic analyses. During the process of transformation of the yeast culture from solid medium to liquid Inoculum I and II, the samples are always analyzed using microscopy. Where both the purity of the culture (potential contamination) and the yeast cells are checked. At the beginning of processing the production medium, the culture is again checked for possible contamination.

The next step, when it is possible to identify contamination or the presence of other types of microorganisms, is the HPLC analysis of lipid substances. The HPLC analysis used enables the identification of the produced sterols. Thus, if a foreign organism, e.g. bacteria, were present in the sample, it would be possible to identify them by the presence of bacterial sterols. At the same time, a certain identification element is the profile of the carotenoids produced.

Regarding the atom economy. Screening experiments and laboratory bioreactor cultivations were pilot experiments. Our goal was to test whether the selected carotenogenic yeast strains are able to grow on this combination of substrates and whether the productivity will exceed the control media. From this point of view, we have not yet addressed the complete processing of the material and the yield of individual components in the medium. As part of the optimization of cultivation conditions and the composition of the medium, we will study the content of waste substances in the medium in more detail. Which also corresponds with our intention of applying the principles of the circular economy, where the goal is to process the maximum amount of used substrates.

Our future idea is also to use fermentation waste as a basis for the preparation of a new medium with an analysis of the composition of individual residual components and its possible use in the following cultivation process.

Changes we made in the manuscript:

We added a table of measured production of metabolites and biomass of Inoculum I of the Rhodosporidium toruloides strain into the supplementary file. This inoculum was subsequently used for inoculation of Inoculum II and fermentor cultivation.

Reviewer 2 Report

In my opinion the paper decribes very promising method of pigemnt fermenting yeast production with use of waste raw-materials. The manuscript contains knowledge with high novelty ratio. It is worth to be published . To improve teh manuscript I suggest only to add two things in the "methods" part: In lines 12-15 it should be more explained why doubled inoculation was performed and in line 127 it should be added apropriate reference number decsribing Folch's method of extraction, this also propriate refernece should be aded to the references part. 

Author Response

Reviewer #2

RE: „ Jiří Holub et al.: „Production of enriched biomass by carotenogenic yeasts cultivated on by-products of poultry processing – a screening study.“   manuscript-revised.

Foremost, we would like to thank you for your great efforts in providing constructive criticism. We have carried out this revision according to the comments made by you as outlined below. All changes are indicated in Green within the manuscript. For these changes page or line numbers are also included.

Reviewer #1 Comments and Suggestions for Authors

In my opinion the paper decribes very promising method of pigemnt fermenting yeast production with use of waste raw-materials. The manuscript contains knowledge with high novelty ratio. It is worth to be published . To improve teh manuscript I suggest only to add two things in the "methods" part: In lines 12-15 it should be more explained why doubled inoculation was performed and in line 127 it should be added apropriate reference number decsribing Folch's method of extraction, this also propriate refernece should be aded to the references part.

Our comments:

The materials and methods section has been modified. The reason why we chose the method of using double inoculation was added. We are based on our many years of experience working with carotenogenic yeasts. When the procedure of this double inoculation ensures that for the inoculation of the production medium the culture is in the exponential phase of growth and the optical density of the culture is higher and therefore the number of viable cells.

Changes in the manuscript:

Yeast cultivation was carried out by double inoculation in Erlenmeyer flasks in YPD media (Tab. 4), according to an inoculation ratio of 1:5, on reciprocal shakers ensuring constant shaking. Double inoculation process was used based on experience and previous published results [12]. This inoculation scheme is sufficient enough to produce culture with high cell density and in exponencial growth phase. Each inoculum was re-inoculated 24 h after inoculation. Inoculation into production media from the 2nd inoculum was performed at an inoculation ratio of 1:5 to 50 ml of the production media in 250 ml Erlenmeyer flasks. Production cultivation lasted 96 h at constant shaking on reciprocating shakers. Cultivation was terminated by separating the biomass from medium by repeated centrifugation and subsequent freezing of the biomass at -80 °C to prepare it for the lyophilization procedure.

The reference number for original Folch extraction method was added to the manuscript also wifh a reference numbers of our adapted Folch extraction method for carotenogenic yeasts lipid metabolites analysis. See section  2.6      Gravimetry and pigment extraction

The references are:

  • SZOTKOWSKI, M.; HOLUB, J.; ŠIMANSKÝ, S.; HUBAČOVÁ,K.; HLADKÁ, D.;  NEMCOVA, A.; MAROVA, I.. Production of Enriched Sporidiobolus sp. Yeast Biomass Cultivated on Mixed Coffee Hydrolyzate and Fat/Oil Waste Materials. Microorganisms. 2021, 9(9). DOI:10.3390/microorganisms9091848
  • FOLCH, J.; LEES, M.; SLOANE STANLEY, G. H. A simple method for the isolation and purification of total lipids from animal tissues. J biol Chem, 1957, 226(1), 497-509.
  • Szotkowski, M.; Byrtusova, D.; Haronikova, A.; Vysoka, M.; Rapta, M.; Shapaval, V.; Marova, I. Study of metabolic adaptation of red yeast to waste animal fat substrate. Microorganisms 2019, 7, 578.

Round 2

Reviewer 1 Report

accept